# Correlation of Matrisome-Associatted Gene Expressions with LOX Family Members in Astrocytomas Stratified by IDH Mutation Status

**DOI:** 10.3390/ijms23179507

**Published:** 2022-08-23

**Authors:** Talita de Sousa Laurentino, Roseli da Silva Soares, Suely Kazue Nagahashi Marie, Sueli Mieko Oba-Shinjo

**Affiliations:** Cellular and Molecular Biology Laboratory (LIM 15), Neurology Department, Faculdade de Medicina (FMUSP), Universidade de Sao Paulo, Sao Paulo 01246-903, SP, Brazil

**Keywords:** lysyl oxidase, matrisome, glioblastoma, diffuse astrocytic, progression, LOX, LOXL1, LOXL3, extracellular matrix

## Abstract

Tumor cell infiltrative ability into surrounding brain tissue is a characteristic of diffusely infiltrative astrocytoma and is strongly associated with extracellular matrix (ECM) stiffness. Collagens are the most abundant ECM scaffolding proteins and contribute to matrix organization and stiffness. LOX family members, copper-dependent amine oxidases, participate in the collagen and elastin crosslinking that determine ECM tensile strength. Common IDH mutations in lower-grade gliomas (LGG) impact prognosis and have been associated with ECM stiffness. We analyzed the expression levels of LOX family members and matrisome-associated genes in astrocytoma stratified by malignancy grade and IDH mutation status. A progressive increase in expression of all five LOX family members according to malignancy grade was found. *LOX*, *LOXL1*, and *LOXL3* expression correlated with matrisome gene expressions. LOXL1 correlations were detected in LGG with *IDH* mutation (IDH^mut^), *LOXL3* correlations in LGG with IDH wild type (IDH^wt^) and strong *LOX* correlations in glioblastoma (GBM) were found. These increasing correlations may explain the increment of ECM stiffness and tumor aggressiveness from LGG-IDH^mut^ and LGG-IDH^wt^ through to GBM. The expression of the mechanosensitive transcription factor, β-catenin, also increased with malignancy grade and was correlated with *LOXL1* and LOXL3 expression, suggesting involvement of this factor in the outside–in signaling pathway.

## 1. Introduction

Invasiveness, the infiltrative capacity of tumor cells into surrounding tissue, is a major characteristic of gliomas that prevents complete tumor resection, leads to inexorable tumor recurrence, and confers poor clinical outcome. Tumor extracellular matrix (ECM) stiffness has emerged as a physical hallmark of cancer that, contributes to cancer initiation, progression, metastasis and metabolic reprogramming [1,2], and resistance to drug and radiation therapy [3]. The ECM, also known as the matrisome, is composed of a complex cross-linked meshwork of over 1000 core matrisomes, including ECM glycoproteins, proteoglycans, and collagens, besides matrisome-associated proteins, including ECM-affiliated proteins, ECM regulators, and secreted factors. Collagens are the most abundant ECM scaffolding proteins and contribute to ECM organization and stiffness through crosslinking mediated by lysyl oxidase (LOX) family members [4,5]. This family comprises five cooper-dependent amine oxidases (LOX, LOXL1, LOXL2, LOXL3, and LOXL4) [6,7], which are involved in several hallmarks of cancers [8,9,10], such as tumor microenvironment remodeling, invasion/migration [11,12,13], growth [14,15,16], inflammatory response [17,18,19], genomic stability [20], and resistance to chemotherapy [21,22].

Astrocytomas, or astrocytic gliomas, are the most common brain tumor. These tumors are classified according to their histologic and molecular characteristics and, more recently, according to isocitrate dehydrogenase (IDH) mutation status, where the presence of this mutation confers a more favorable prognosis [23]. Low-grade astrocytic gliomas (LGG), encompassing grades 2 and 3, are more well-differentiated, slower-growing tumors than grade 4 glioblastoma (GBM), with a median overall survival (OS) of 15 months [24]. Therefore, tumor aggressiveness progressively increases from LGG with IDH mutation (LGG-IDH^mut^) to LGG with IDH wild type (LGG-IDH^wt^), through to GBM. Tumor recurrence occurs independently of other factors, but time to recurrence varies according to malignancy grade and IDH mutational status, proving shorter in IDH^wt^ than IDH^mut^ tumors [25]. A growing body of evidence indicates that denser ECM leads to more aggressive tumor progression. Recently, an association between ECM stiffness oscillation and IDH mutation status was established [26].

Our group has previously reported a correlation between increased LOX expression and higher malignancy grade in human astrocytomas, and also a correlation of LOX expression with *IDH1* mutation status [27]. In the present study, the expression of the other members of the LOX family in astrocytomas, stratified by malignancy grade and IDH mutation status, was explored. The differentially expressed LOX member genes were subsequently correlated with matrisome-associated gene expression in LGG-IDH^mut^, LGG-IDH^wt^, and GBM groups. Additionally, a search for mechanosensitive transcriptional factors, whose expression correlated with the group of differentially expressed genes, was performed. The study objective was to identify the LOX family members, along with matrisome components, that impact astrocytoma ECM stiffness. This knowledge can help identify LGG-IDH^wt^ that are more prone to tumor progression, together with new candidates for interventions aimed at reducing tumor invasiveness.

## 2. Results

### 2.1. LOX Family Expression Levels in Different Malignant Grades of Astrocytomas

Gene expression analysis by quantitative real-time PCR (RT-qPCR) for *LOX*, *LOXL1*, *LOXL2*, *LOXL3*, and *LOXL4* showed increased expression levels in astrocytoma samples compared to non-neoplastic (NN) samples. Moreover, these expressions increased with malignancy, proving the highest in GBM. Pairwise comparisons of NN samples and diffusely infiltrative astrocytoma grades 2, 3, and 4 (AG2, AG3, and GBM, respectively) were significant (*p* < 0.0001) for all five LOX family genes. The paired comparisons relative to GBM were also significant, except for the LOXL3-AG3 comparison (Figure 1A). The results for this cohort were confirmed in an independent larger cohort of The Cancer Genome Atlas (TCGA) and Genotype-Tissue Expression (GTEx), showing significant differences among groups for the five genes analyzed (*p* < 0.0001), with the highest significance found in the paired comparisons with GBM (Figure 1B). Expression profiles of *LOX*, *LOXL1*, *LOXL2*, *LOXL3*, and *LOXL4* in the TCGA cohort were also evaluated in low grade astrocytic glioma (LGG), encompassing AG2 and AG3, according to IDH mutation status and for GBM cases (Figure 1C). Gene expression levels were higher in LGG-IDH^wt^ than in LGG-IDH^mut^ for all genes except *LOXL2*. Comparisons for the LGG-IDH^wt^-GBM pair showed significant differences for all members of the LOX family, with higher expression in GBM, except for LOXL4. Significant *LOX*, *LOXL1*, and *LOXL3* differential expressions were observed in both LGG-IDH^mut^ vs. LGG-IDH^wt^ and LGG-IDH^wt^ vs. GBM comparisons. Therefore, these three genes were selected for further analysis.

### 2.2. LOX, LOXL1, and LOXL3 Protein Expression Analyses and Gene Expression Impact on Prognosis

Protein expressions of LOX, LOXL1, and LOXL3 were investigated by immunohistochemistry in LGG-IDH^mut^, LGG-IDH^wt^, GBM-IDH^mut^, and GBM-IDH^wt^. A progressive increase in expression of these proteins was observed from LGG-IDH^mut^, LGG-IDH^wt^, and through to GBM, proving highest in GBM and confirming the results of the gene expression analyses (Figure 2).

The impacts of *LOX*, *LOXL1*, and *LOXL3* expression levels were analyzed for OS in LGG and GBM, based on the TCGA database. In LGG, patients with higher levels of LOX (*p* = 0.033) and *LOXL1* (*p* = 0.002) expression had shorter OS than patients with lower expression (Figure 3A). LGG with highest and lowest levels of *LOX* expression had OS of 49.91 ± 7.38 months and 57.89 ± 10.59 months (*p* = 0.033), respectively. LGG with the highest and lowest levels of *LOXL1* expression had OS of 19.87 ± 5.25 months and 57.89 ± 8.06 months (*p* = 0.0034), respectively. Similarly, OS of GBM patients was analyzed using Kaplan–Meier curves. Only *LOXL1* expression levels impacted the prognosis of GBM, where patients with high expression had shorter OS (Figure 3B). GBM patients with *LOXL1* overexpression had mean survival of 10.28 ± 1.11 months, while those with lower expression had mean survival of 13.76 ± 1.09 months (*p* = 0.0034). Multivariate Cox regression analysis with age at diagnosis identified only *LOXL1* expression as an independent variable for predicting prognosis in LGG (*p* = 0.027) (Figure 3C) and GBM patients (*p* = 0.032) (Figure 3D). In addition, radiotherapy was indicated to 45 (31.5%) out of 143 LGG patients presenting low *LOXL1* expression, in contrast to 55 (49.5%) indications among 111 LGG patients with high *LOXL1* expression. Nevertheless, the survival time was shorter for those with high *LOXL1* expression (Figure 3E and Appendix A).

### 2.3. Matrisome Analysis

An in silico analysis of LOX family genes and of genes coding for matrisome proteins was performed to evaluate correlations among the expression profiles using the TCGA and GTEx databases, stratifying cases into LGG and GBM, according to IDH mutation status. Initially, the expression levels of the genes coding for the 1027 matrisome proteins and the of LOX family genes in LGG and GBM compared to NN and LGG compared to GBM, were compared using Limma-voom, as per the workflow depicted in Figure 4. Genes with a log fold change (FC) > |1| and an adjusted *p* ≤ 0.05 in any of the comparisons were selected, resulting in 439 upregulated genes and 233 downregulated genes, including *LOX*, *LOXL1*, and *LOXL3*. Expression of *LOXL2* and *LOXL4* did no differ significantly on any of the comparisons. Subsequently, a total of 672 up and downregulated genes in LGG-IDH^mut^, LGG-IDH^wt^, and GBM groups were analyzed using Kruskal–Wallis and Dunn’s post hoc tests (LGG-IDH^wt^ vs. LGG-IDH^mut^ and LGG-IDH^wt^ vs. GBM). This analysis led to the identification of 70 differentially expressed genes, comprising 9 genes coding for collagens, 17 for ECM glycoproteins, 23 for ECM regulators, 5 for ECM-affiliated proteins, 3 for proteoglycans, 10 for secreted factors, and 3 LOX genes (LOX, LOXL1, and LOXL3) (Appendix A). The workflow of the selection is presented in Figure 4. The heat map of these 70 genes is presented in Figure 5A. Higher gene expression levels were observed in LGG-IDH^wt^ than in LGG-IDH^mut^, whereas the highest expression levels were detected in GBM cases, particularly in the mesenchymal molecular subtype.

The strength of the correlations of each of the three LOX genes with the 67 matrisome-related genes was then determined by Pearson’s correlation test (Appendix A). In LGG-IDH^mut^, 22 genes exhibited a moderate correlation (0.7 < R < 0.4, *p* < 0.05), 1 gene, *CTHRC1*, had strong correlation (R = 0.7, *p* < 0.0001) with *LOXL1*, and only 2 genes were moderately correlated with *LOXL3*. None of the genes showed a significant correlation with *LOX*. In LGG-IDH^wt^, 15, 21, and 2 genes had moderate correlations with *LOXL1*, *LOXL3*, and *LOX*, respectively, while *S100A11* and *CTSB* showed strong correlations with *LOXL3* (R = 0.751, *p* < 0.001 and R = 0.740, *p* < 0.001, respectively). In GBM, 28, 28, and 43 genes showed moderate correlations with LOXL1, *LOXL3*, and *LOX*, respectively, while *SERPINE1* and *PLOD2* displayed strong correlations with *LOX* (R = 0.707, *p* < 0.001 and R = 0.702, *p* < 0.001, respectively). The levels of expression of the matrisome-associated genes correlated (in increasing complexity) with *LOXL1* in LGG-IDH^mut^, *LOXL3* in LGG-IDH^wt^ and with the three LOXs, but most strongly with *LOX* in GBM (Figure 5B).

Potential involvement of mechanosensitive transcription factors (TFs) (*CTNNB1*, *HIF1A*, *JUN*, *JUNB*, *LEF1*, *NFKB1*, *SMAD2*, *SMAD3*, *SNAI1*, *SOX2*, *STAT3*, *TWIST1*, *YAP1*) in the observed expression profiles was investigated by analyzing the correlation of these TFs with *LOXL1*, *LOXL3*, and *LOX* expressions. *NFKB1* and *SOX2* were differentially expressed across the groups analyzed (LGG-IDH^mut^, LGG-IDH^wt^, and GBM) but correlated only with *LOXL3* and *LOX* expression levels in LGG-IDH^wt^ and GBM, respectively (data not shown). Interestingly, the level of *CTNNB1* expression was significantly higher in LGG-IDH^wt^ than in LGG-IDH^mut^ (*p* < 0.0001), and also higher in GBM related to LGG-IDH^wt^ (*p* = 0.0081) (Figure 6A), correlating with *LOXL1* and *LOXL3* expression both in LGG-IDH^mut^ and LGG-IDH^wt^ (Figure 6B).

## 3. Discussion

ECM stiffness contributes significantly to cancer progression by promoting tumor cell proliferation, invasion, and drug resistance. In fact, the matrix stiffness of glioma is higher than that of normal brain tissue and increases with malignancy [28,29]. The LOX family is widely known as amine oxidase enzymes responsible for crosslinking of ECM collagens and/or elastin that determines ECM tensile strength, remodeling, and integrity [6,7,18,30,31,32,33]. LOX is the most extensively studied member of this family and, in a previous study by our group, a progressive increase in LOX expression according to malignancy was observed in human astrocytomas [34]. In the present study, the role of the other LOX family members in the progression of astrocytomas was confirmed. *LOXL1*, *LOXL2*, *LOXL3*, and *LOXL4* expression levels were also found to increase progressively in astrocytomas from grades 2 to 4, proving highest in GBM, and significantly higher relative to non-neoplastic brain tissue. These results were subsequently confirmed in silico for the TCGA glioma RNAseq dataset. Similar results have been observed for other tumors, suggesting a key role of the LOX family members in tumor progression [9,18,21,35,36,37,38,39,40]. In a previous study by our group, a correlation of LOX expression level with IDH1 mutation status was found in diffusely infiltrative astrocytomas (grades 2 to 4) [27]. Similarly, significant differential expression of the other LOX family members according to IDH mutational status was also observed in the present investigation. Oncogenic IDH mutations lead to decreased levels of α-ketoglutarate (α-KG) and accumulation of 2-hydroxyglutarate (2HG) [41], an oncometabolite linked to increased reactive oxygen species (ROS) production [42] and decreased mitochondrial respiration in GBM cells [43]. Moreover, 2HG is a competitive inhibitor of α-KG dependent dioxygenases, including histone demethylases and the ten eleven translocation (TET) family of 5-methylcytosine (5mC) hydroxylases, which promote marked epigenetic alterations [44]. Given α-KG is also required by prolyl hydroxylases to promote the degradation of hypoxia-inducible factor 1 alpha (HIF-1α), the IDH mutation with decreased α-KG level leads to HIF-1α stability [45]. Thus, increased HIF-1α promotes ECM remodeling through regulation of collagen deposition by tumor cells [46] and upregulation of metalloproteinases and collagen-modifying enzymes in stromal cells [47]. Glioma aggressiveness and patient outcomes have also been found to correlate with HIF-1α levels and tenascin C-enriched ECM stiffness. IDH^mut^ has been shown to restrict tumor aggression by decreasing HIF1α-dependent tenascin C expression, thereby decreasing ECM stiffness and mechanosignaling [28]. In fact, ECM stiffness has been described as significantly lower in IDH^mut^ than in IDH^wt^ tumors [28] and, clinically, LGG-IDH^wt^ has a poorer prognosis than LGG-IDHmut [48]. Moreover, reports show *LOX* [49], *LOXL1*, and *LOXL3* [50] expression levels are modulated by HIF-1α. Regulation of ECM stiffness and glioma cell migration by LOX expression have been shown in drosophila and mouse models [51], and LOX activity was reduced by HIF-1α knockdown [50]. Correlations of *LOX* expression with epithelial mesenchymal transition (EMT) and IDH1 mutation status have also been described [52]. Similarly, *LOXL1* has been associated with tumor invasion, metastasis, and extracellular accumulation of lactate [53], linked to integrin α11, a stromal collagen receptor [54]. Associations of *LOXL3* expression with EMT (through E-cadherin transcription repression by SNAIL) [55,56], cell invasion in breast cancer [57], and with GBM cell adhesion and invasion [10] have been described. Interestingly, genes related to EMT were overexpressed in IDH^wt^ [58].

In the present study, a progressive increase in levels of *LOX*, *LOXL1*, and *LOXL3* expression was observed for LGG-IDH^mut^, LGG-IDH^wt^, and GBM in the TCGA glioma RNAseq dataset, suggesting their role in determining ECM composition and stiffness enhancement in these phenotypes.

ECM stiffness is related to matrix composition, matrix contraction, and matrix crosslinking, where LOX family members play a major role [59]. Concerning matrix composition, collagen is an ECM scaffolding protein that contributes significantly to the tensile strength of tissue, binding cells by forming specialized extracellular networks [60]. Fibrillar collagen types I-III, V, and XI are the most common [61], and LOX, LOXL1, and LOXL3 are preferentially associated with fibrillar collagen types I and III [6]. In this study, *COL1A1*, *COL1A2*, *COL3A1*, and *COL5A1* expression levels were correlated with all three LOXs in GBM, whereas *COL6A1-3* expression levels correlated with *LOXL1* and *LOXL3* in LGG, and with all three LOXs in GBM. Fibrillar collagen deposition was described in the adventitia of remodeled large vessels and glomeruloid vascular structures of GBM [62]. High *COL6A1* expression levels were observed in astrocytoma of different malignancy grades, especially in higher grade tumors associated with poor prognosis [63]. COL6A1 was detected in perivascular regions and pseudopalisading cells, while COL6A1 expression was associated with hypoxia [64] and VEGF expression [65].

Results showed overexpression of two genes coding for lysyl hydroxylases, procollagen-lysine, and 2-oxoglutarate 5-dioxygenase (PLOD) [66], *PLOD1*, and *PLOD2*, involved in collagen biosynthesis and crosslinking [66,67] in LGG-IDH1^wt^ and GBM. Similar results were recently reported by other authors [68,69] and in other cancer types with aggressive phenotype [70]. We also found upregulation of another gene related to collagen biosynthesis, the serpin family H member 1 (*SERPINH1*), which was significantly correlated with LOXL1 in LGG-IDH^wt^ and GBM, and with LOX in GBM. *SERPINH1* encodes a heat shock protein (HSP47) localized in the endoplasmic reticulum, participates in the correct folding of collagen [71], and facilitates its secretion and deposition [72]. Increased expression of HSP47 has been associated with high malignancy grade of glioma [73]. Transforming growth factor β1 (TGF-β1) is also an inducer of collagen biosynthesis [74], and the gene coding for the transforming growth factor beta induced (TGFBI) was overexpressed and correlated significantly with LOX expression in GBM. TGFBI binds to collagens I, II, and IV and, therefore, mediates cell–collagen interaction. TGFBI also inhibits cell adhesion, promotes cell migration in glioma cells [75] and has been associated with the expression signature of mesenchymal GBM [76], the molecular subtype with the poorest prognosis.

The organization of collagen fibrils, including modulation of their diameters and interfibrillar spacing [77], is performed by lumican (coded by *LUM*), a class II leucine rich proteoglycan. The ECM organization process assembles adhesion plaque complexes and allows integrins to transduce cues from the ECM by activating a signaling cascade that induces cytoskeletal remodeling and regulates cell behavior [78]. Thus, LUM may enhance cancer growth through integrin β1 activating the β-catenin/focal adhesion kinase (FAK) [79]. Fibronectin [80] also induces α5β1 signaling and rapidly activates the downstream cascade through FAK, providing tensile support for motility of cancer cells in the invasion process [81]. In addition, β1 integrins and tumor adhesion to FN1 mediate resistance to radiotherapy [82]. Previous studies have reported that radiotherapy increases LOX secretion [83]. In the current analysis, both *LUM* and *FN1* were upregulated and showed significant correlation with *LOX* in GBM. A high abundance of FN1 protein in the ECM of GBM was previously demonstrated by our group [84].

Metabolically, IDH^wt^ possesses a cytosolic substrate composition that is better suited for collagen biosynthesis. Proline (Pro) constitutes about 10% of total amino acids in collagen [85], and can be synthesized from arginine, glutamine (Gln), and glutamate (Glu) [86]. The presence of IDH^mut^ in glioma leads to a significant reduction in Gln and Glu levels, whereas IDH^wt^ gliomas contain high levels of intracellular Glu [87,88], providing Pro for collagen biosynthesis through the activity of two enzymes: aldehyde dehydrogenase 18 family member A1, coded by *ALDH18A1*, and pyrroline-5-carbohydrate reductase 1, coded by *PYCR1*. These two enzymes synthesize Glu to glutamate ϒ-semialdehyde and then to Pro, respectively. Interestingly, both these genes were upregulated in both LGG and GBM (data not shown).

In addition to Glu accumulation, an increase in lactate level can be observed in IDH^wt^ due to the Warburg effect, with a shift of the oxidative tricarboxylic acid cycle towards glycolysis [89]. Glycolytically derived acids are transported to the extracellular microenvironment by membrane ion pumps and transporters, substantially lowering extracellular pH [90]. The acidification of tumoral environments produces favorable conditions for tumor cell invasion, as it induces the formation and maturation of invadopodia and activates proteases to focally degrade the ECM [91], through metalloproteinase 3 activity [92] and MMP9 secretion [93]. Moreover, acid-activated protease, such as cathepsin B (CTSB) [94], is secreted into the ECM at low pH. CTSB is located predominantly in secretory vesicles [95] and participates in ECM remodeling by degrading ECM components such as collagen, laminin, fibrin, elastin and TNC [96,97,98]. In gliomas, CTSB can bind to ANXA2 and induce the expression of vascular endothelial growth factor C, TGF-β, and MMP9 to promote angiogenesis [99,100]. *CSTB* upregulation has been described in GBM [101] and GBM stem cells [102], particularly in IDH^wt^, and has been strongly associated with the mesenchymal subtype and immunosuppressive conditions in gliomas [94]. Interestingly, *CTSB* was highly correlated with *LOXL3* in LGG-IDH^wt^ and with LOX in GBM. The 2-HG generated in IDH^mut^ enhances angiogenesis through HIF1α stabilization, partially by decreasing levels of endostatin, an HIF1α antagonist, which in turn increases vascular VEGF signaling [103]. VEGFA was upregulated and significantly correlated with *LOX* in GBM. Additionally, the expression level of the collagen triple helix repeat containing-1 gene (*CTHRC1*), an ECM glycoprotein inducer of angiogenesis, migration, and cell invasion, showed the highest correlation with LOXL1 in LGG-IDH^mut^. FN1, significantly correlated with LOX expression in GBM in the present analysis, may also promote angiogenesis in cancer by providing a ridged structure for vessel development and signaling for endothelial cell migration [104].

Lastly, the ways in which ECM stiffness can trigger mechanotransduction were analyzed by searching for correlations between the differentially expressed genes and mechanosensitive transcription factors. Interestingly, there was a progressive increase in *CTNNB1*, which codes for β-catenin, according to astrocytoma malignancy grade, and its expression correlated significantly with *LOXL1* and *LOXL3* expression in both LGG-IDH^mut^ and LGG-IDH^wt^. Collagen deposition and crosslinking promote ECM stiffness and clustering of integrins, leading to FAK activation with subsequent PI3K/AKT activation, inhibition of GSK-3β, and stabilization of β-catenin [105]. Nuclear accumulation of β-catenin and induction of stemness by matrix stiffness was observed in glioma cells [29]. LOXL1 has previously been shown to regulate cell migration and apoptosis via Wnt/β-catenin signaling [106]. However, this is the first report of correlation of *LOXL3* with β-catenin expression level in cancer. Recently, our group identified upregulation of LOXL1 and involvement of Wnt/β-catenin signaling in the malignant transformation of normal astrocytes by anoikis [107]. We speculate that *LOXL1*, and its interactions with the matrisome-related genes, may be involved in this first step of transformation towards an astrocytic tumor. Additionally, elements of TCF/LEF transcription factors were identified in the promoter region of LOXL1, suggesting that the β-catenin signaling pathway may regulate positively LOXL1 expression [108].

Overall, the results showed a significant correlation of increased gene expression of *LOXL1*, *LOXL3* and *LOX* with matrisome-related gene expression, possibly explaining the progressive increase in ECM stiffness from LGG-IDH^wt^, LGG-IDH^wt^, and through to GBM. Figure 7 summarizes our findings. More specifically, the *LOXL1* correlation net was detected in less malignant astrocytomas. Connectivity with *LOXL3* was prominent among more aggressive LGG-IDH^wt^, and strong connectivity with *LOX* was detected in GBM, demonstrating the complexity of ECM components involved in astrocytoma progression. More specifically, level of LOXL1 expression impacted the outcome of LGG and GBM patients, with shorter OS in individuals exhibiting higher expression. In addition, radiotherapy was indicated more frequently in LGG patients with high LOXL1 expression, yet this group had shorter survival than LGG patients with low LOXL1 expression.

## 4. Materials and Methods

### 4.1. Tissue Samples

The casuistic samples consisted of 130 diffusely infiltrative astrocytomas (grades II to IV). Tumors were graded according to the WHO classification AG2 (n = 26, mean age at diagnosis 34.0 ± 8.1 years, 15 males and 11 females), AG3 (n = 18, mean age at diagnosis 35.0 ± 12.3 years, 11 males and 7 females), and GBM (n = 86 mean age at diagnosis 54.0 ± 13.9 years, 58 males and 28 females). The non-neoplastic control group consisted of samples from individuals undergoing temporal lobe resection during epilepsy surgery (n = 22, mean age at diagnosis 38.0 ± 7.6 years, 10 males and 12 females). All samples were collected during surgical procedures by the Neurosurgery Group of the Department of Neurology at the Hospital das Clinicas of the School of Medicine of University of Sao Paulo. Fresh surgical samples were immediately snap-frozen in liquid nitrogen upon surgical removal. Before RNA extraction, a 4-μm-thick section of each sample was obtained for histological assessment using a cryostat at −25 °C. Sections were stained with hematoxylin and eosin and examined under light microscope. Necrotic and non-neoplastic areas were removed from the frozen block of tumoral tissue by microdissection prior to RNA extraction. Grey matter was avoided in the control samples. Written informed consent was obtained from all patients according to the ethical guidelines approved by the Ethical Committee of the School of Medicine, University of São Paulo (0599/10).

### 4.2. Total RNA Extraction and cDNA Synthesis

Total RNA was extracted from frozen tissues using the RNeasy Mini kit (Qiagen, Hilden, Germany). RNA concentration and purity were determined by measuring absorbance at 260 and 280 nm. Ratios of 260/280 measures ranging from 1.8 to 2.0 were considered acceptable for purity standards. Denaturing agarose gel electrophoresis was used to evaluate the integrity of samples. A conventional reverse transcription reaction was performed to yield single-stranded cDNA. The first strand of cDNA was synthesized from 1000 ng of total RNA previously treated with one unit of DNase I (FPLC-pure, GE Healthcare, Uppsala, Sweden) using random and oligo (dT) primers, RNase inhibitor, and SuperScript III reverse transcriptase according to the recommendations of the manufacturer (Thermo Fisher Scientific, Carlsbad, CA, USA). The resulting cDNA was subsequently treated with one unit of RNase H (GE Healthcare), diluted with TE buffer, and stored at −20 °C until later use.

### 4.3. Reverse Transcription Quantitative Real Time PCR

Analysis of relative expression levels of LOXL1, LOXL2, LOXL3, and LOXL4 were performed by RT-qPCR using the Sybr Green I approach. Quantitative data were normalized in relation to the geometric mean of three reference genes: glucuronidase beta (GUSB), hypoxanthine phosphoribosyltransferase (HPRT), and TATA box binding protein (TBP). Table 1 shows the specific primer sequences and concentrations used for RT-qPCR. The minimum primer concentrations necessary were determined to give the lowest quantification cycle (Cq) and maximum amplification efficiency, while minimizing non-specific amplification. Primer concentrations used were 200 nM for all primers, except for GUSB which was 400 nM. Standard curves were established to ensure amplification efficiency, and analysis of melting curves demonstrated a single peak for all PCR products. Additionally, agarose gel electrophoresis was employed to check the size of the PCR products amplified. Sybr Green I amplification mixtures (12 μL) contained 3 μL of cDNA, 6 μL of Power Sybr Green I Master Mix (Thermo Fisher Scientific), and forward and reverse primers. PCR reactions were run on an ABI Prism 7500 sequence detector (Thermo Fisher Scientific) as follows: 2 min at 50 °C, 10 min at 95 °C, and 40 cycles of 15 s at 95 °C, and 1 min at 60 °C. The equation 2-μCt was applied to calculate the expression in all samples, where ∆Ct = [Ct of gene − geometric mean Cq of reference genes]. The RT-qPCR reactions for each sample were performed in duplicates and repeated when the Cq values were not similar. The results are presented on a log10 scale for better visualization. The gene expression levels were scored according to the median relative expression values of each astrocytoma grade. For statistical analysis, scores equal or higher than the median values were defined as overexpression.

### 4.4. Immunohistochemistry

The immunohistochemical procedures for LOX, LOXL1, and LOXL3 expression analyses were performed on 4-μm slices of paraffin-embedded tissues of 6 cases each of AG2, AG3, and GBM. Tissue sections were first subjected to antigen retrieval in 10 mM citrate buffer, pH 6.0, and incubated at 122 °C for 3 min using an electric Pascal (BioCare Medical, Walnut Creek, CA, USA). Specimens were then blocked and further incubated with primary antibodies against human LOX (ab31238, Abcam, Cambridge, UK), LOXL1 (HPA 042111, Sigma-Aldrich, St Louis, MO, USA), and LOXL3 (ARP60280, Aviva, San Diego, CA, USA) at 16–20 °C for 16 h. Slices were then incubated for endogenous peroxidase blocking (Novolink Polymer Detection System, Novocastra, Newcastle upon Tyne, UK) at 16–20 °C for 16 h. Antibodies localization was visualized using diaminobenzidine and Harris hematoxylin. Table 2 show the positive controls were used for all reactions, as well as the dilutions for each antibody.

### 4.5. TCGA

In silico analysis of gene expression was performed in TCGA database (http://www.cbioportal.org, accessed on 1 May 2022) [109,110]. Gene expression dataset from non-tumoral brain samples was obtained from GTEx project (https://gtexportal.org/, accessed on 1 May 2022) [111]. Data was downloaded and read counts were normalized by DeSeq [112]. Data consisted in 115 of non-neoplastic samples, 194 LGG (136 IDH^mut^, 58 IDH^wt^), and 160 samples of GBM (8 G-CIMP, 29 proneural, 38 classical, and 48 mesenchymal subtype). Matrisome gene lists were based on the Matrisome Project (http://matrisome.org/, accessed on 1 May 2022) [113]. Normalized read counts were converted to a z-score for heat map visualization.

### 4.6. Statistical Analysis

The distribution of gene expression data was analyzed by the normality test of Kolmogorov–Smirnov and Shapiro–Wilk test. For gene expression, non-parametric Kruskal–Wallis test was used to compare multiple groups, followed by Dunn’s post-hoc test. Kaplan–Meier survival curves were analyzed using the log-rank test. Multivariate Cox proportional regression analysis was performed using age as a covariate with gene expression level. Receiver Operating Characteristic curves were used to determine the high and low expression groups to Kaplan–Meier analysis for OS of LGG and GBM cases. The cut off of the expression values for ROC analysis was determined for the LGG cases were divided into IDH^mut^ and IDH^wt^, and for GBM samples, the cases were divided into proneural and mesenchymal subtype. Limma package with voom method in R package was used for differentially expressed gene analysis. Correlation analyses between gene expression values were assessed by the non-parametric Pearson’s correlation test. Correlation coefficient R value used was R > 0.5. Statistical significances were considered when *p* ≤ 0.05. All analyses were performed using SPSS version 20.0 (IBM Corporation, Armonk, NY, USA), R software, and plots were made using the program GraphPad Prism version 8.0 (GraphPad Software, San Diego, CA, USA).

## 5. Conclusions

The LOX family was increasingly expressed according to malignancy of astrocytoma, exhibiting the highest expression in GBM. The progressive gene expression connectivity among LOX, LOXL1, and LOXL3 and matrisome-related genes from LGG-IDH^mut^, LGG-IDH^wt^, and through to GBM, reinforce the impact of ECM components contributing for matrix stiffness on the malignant progression and prognosis of astrocytoma.

## Figures and Tables

**Figure 1 ijms-23-09507-f001:**
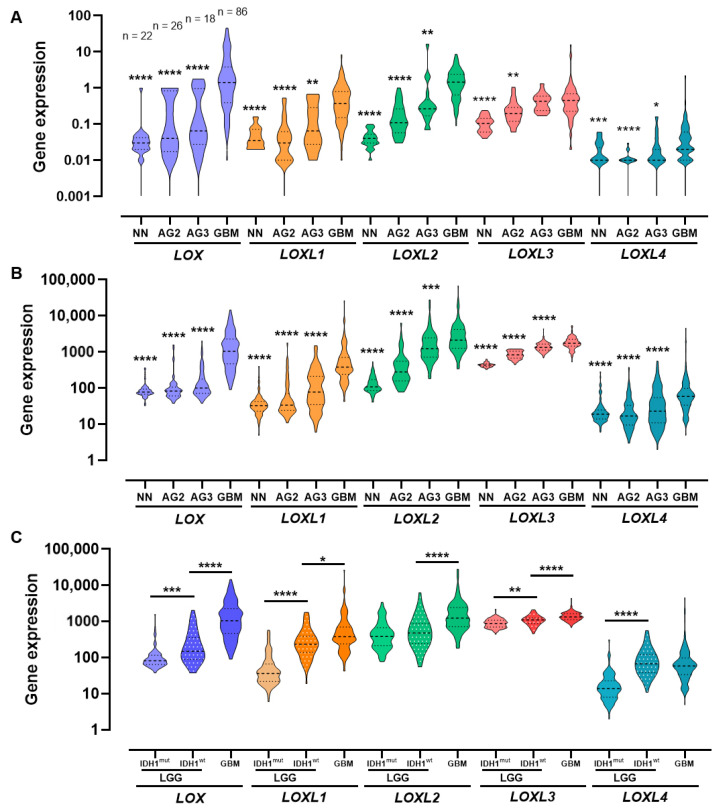
Violin plots showing expression distribution of genes coding for lysyl oxidase family in astrocytomas of different malignancy grades and non-neoplastic brain tissue. (**A**) Expression levels of LOX and LOXL1–4 in cohort determined by RT-qPCR. Statistical analysis was relative to GBM samples. The number of cases in each group is presented in the top of figure. (**B**) Expression levels of *LOX* and *LOXL1–4* in TCGA (astrocytoma groups) and GTEx (NN samples) RNAseq database. Expression increased with malignancy grade of astrocytomas for the four genes analyzed. Statistical analysis was relative to GBM cases. (**C**) Expression levels of *LOX* and *LOXL1–4* in TCGA RNAseq database, for LGG (AG2 and AG3) with (IDH^mut^) and without (IDH^wt^) IDH mutation, and GBM cases. Differences were statistically significant (*p* < 0.0001) as determined by Kruskal–Wallis test for all groups of genes in both cohorts. Middle lines represent median of groups. Top and the bottom lines represent first and third quartiles, respectively. The post-hoc Dunn’s multiple comparison test was used to calculate differences between two groups (* *p* < 0.05; ** *p* < 0.01; *** *p* < 0.001; **** *p* < 0.0001). Abbreviations: AG2, low-grade astrocytoma; AG3, anaplastic astrocytoma; GBM, glioblastoma; GTEx, Genotype-Tissue Expression; IDH^mut^, isocitrate dehydrogenase gene with mutation; IDH^wt^, isocitrate dehydrogenase gene wild type; LGG, lower-grade astrocytic gliomas; NN, non-neoplastic; RT-qPCR, real-time quantitative polymerase chain reaction; TCGA, The Cancer Genome Atlas.

**Figure 2 ijms-23-09507-f002:**
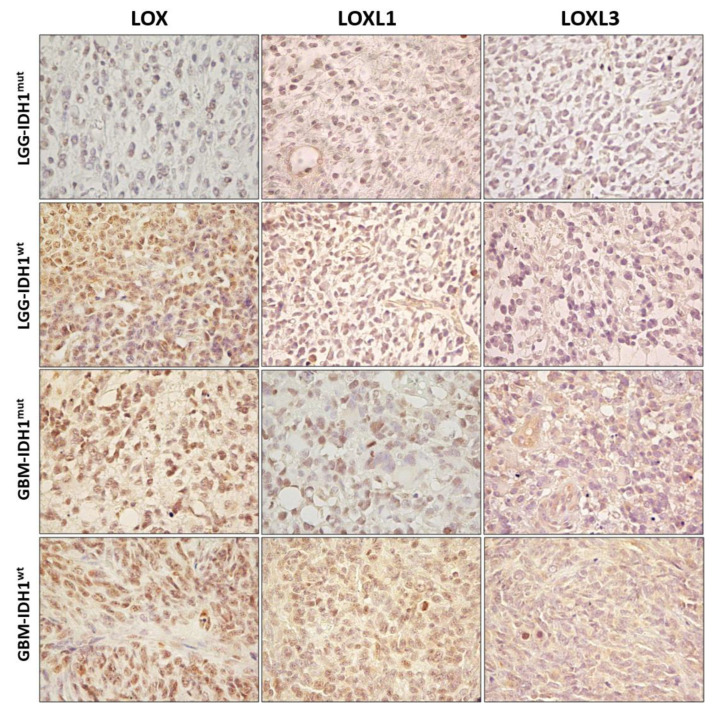
Protein expression levels of LOX, LOXL1, and LOXL3 in astrocytomas. Representative immunohistochemistry slides of lower-grade astrocytic glioma (LGG) and glioblastoma (GBM) with IDH1 mutated (IDH^mut^) and wild type (IDH^wt^). Magnification of 400×.

**Figure 3 ijms-23-09507-f003:**
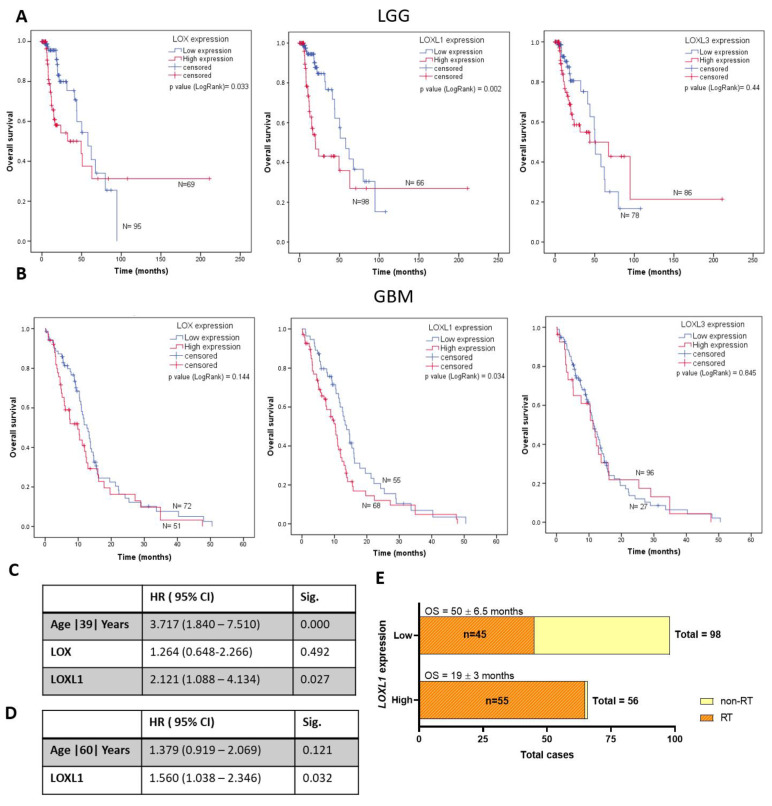
Overall survival curves of LOX family in LGG and GBM cohort in TCGA database. (**A**) Kaplan–Meier curves for overall survival of LGG patients. High- and low-expression groups were determined according to LGG cases with IDH mutation and IDH wild type by ROC curve. (**B**) Kaplan–Meier curve for overall survival of GBM patients. High- and low-expression groups were determined according to GBM cases with proneural and mesenchymal molecular subtypes by ROC curve. (**C**) Multivariate Cox regression of *LOX*, *LOXL1* and age in LGG cases with HR, 95% CI and *p*-values. (**D**) Multivariate Cox regression of *LOXL1* and age in GBM cases with hazard ratios, 95% confidence intervals and *p*-values. (**E**) Overall survival and number of patients submitted or not to radiotherapy with high and low expression of *LOXL1* of LGG cases. Abbreviations: CI, confidence interval; GBM, glioblastoma; HR, hazard ratio; LGG, lower-grade astrocytic gliomas.

**Figure 4 ijms-23-09507-f004:**
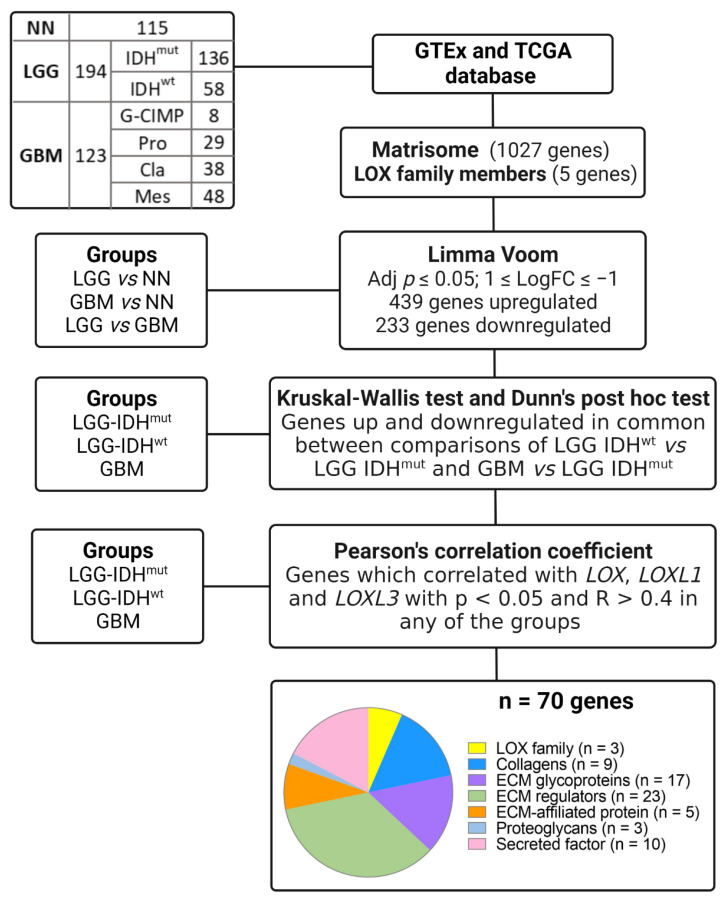
Workflow of in silico analysis of genes coding for matrisome and LOX family in LGG and GBM cohort on TCGA database. Abbreviations: ECM, extracellular matrix; FC, fold change; GBM, glioblastoma; GTEx, Genotype-Tissue Expression; IDH^mut^, isocitrate dehydrogenase gene with mutation; IDH^wt^, isocitrate dehydrogenase gene wild type; LGG, lower-grade astrocytic glioma; NN, non-neoplastic; TCGA, The Cancer Genome Atlas. Created with BioRender.com.

**Figure 5 ijms-23-09507-f005:**
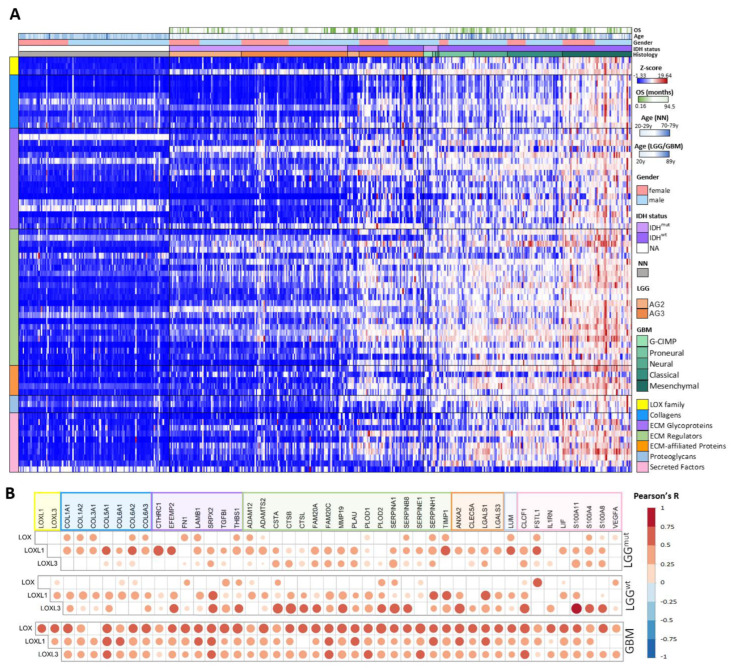
Lysyl oxidase family and matrisome gene expression analyses. (**A**) Heatmap of lysyl oxidase family and matrisome genes in non-neoplastic brain samples and diffuse astrocytomas with IDH mutation status and different subtypes of glioblastoma. The 70 genes selected using the workflow presented in Figure 4 are depicted in the heatmap of z-score of normalized read counts. (**B**) Pearson’s correlation coefficient test of *LOX*, *LOXL1*, and *LOXL3* and matrisome genes in LGG IDH^mut^, LGG IDH^wt^, GBM groups. Abbreviations: AG2, low-grade astrocytoma; AG3, anaplastic astrocytoma; ECM, extracellular matrix; GBM, glioblastoma; IDH^mut^, isocitrate dehydrogenase gene with mutation; IDH^wt^, isocitrate dehydrogenase gene wild type; LGG, lower-grade astrocytic glioma; NA, not analyzed.

**Figure 6 ijms-23-09507-f006:**
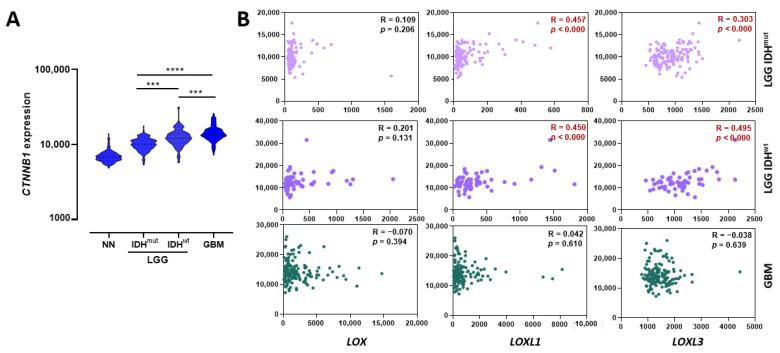
β-catenin gene expression in astrocytoma and relationship with LOX members. (**A**) Expression levels of *CTNNB1* in non-neoplastic tissues, LGG (AG2 and AG3) with (IDH^mut^) and without (IDH^wt^) IDH mutation, and GBM cases. Differences were statistically significant (*p* < 0.0001) as determined by Kruskal–Wallis test. Middle lines represent median of groups. Top and bottom lines represent first and third quartiles, respectively. The post-hoc Dunn’s multiple comparison test was used to calculate differences in expression between two groups (*** *p* < 0.001; **** *p* < 0.0001). (**B**) Pearson’s correlation coefficient test of *CTNNB1* expression levels and *LOX*, *LOXL1*, and *LOXL3* in groups of LGG-IDH^mut^, LGG-IDH^wt^, and GBM. Significative correlations (in red) between *LOXL1* and *LOXL3* with *CTNNB1* were observed in LGG-IDH^mut^ and LGG-IDH^wt^. Abbreviations: GBM, glioblastoma; IDH^mut^, isocitrate dehydrogenase gene with mutation; IDH^wt^, isocitrate dehydrogenase gene wild type; LGG, lower-grade astrocytic glioma; NN, non-neoplastic.

**Figure 7 ijms-23-09507-f007:**
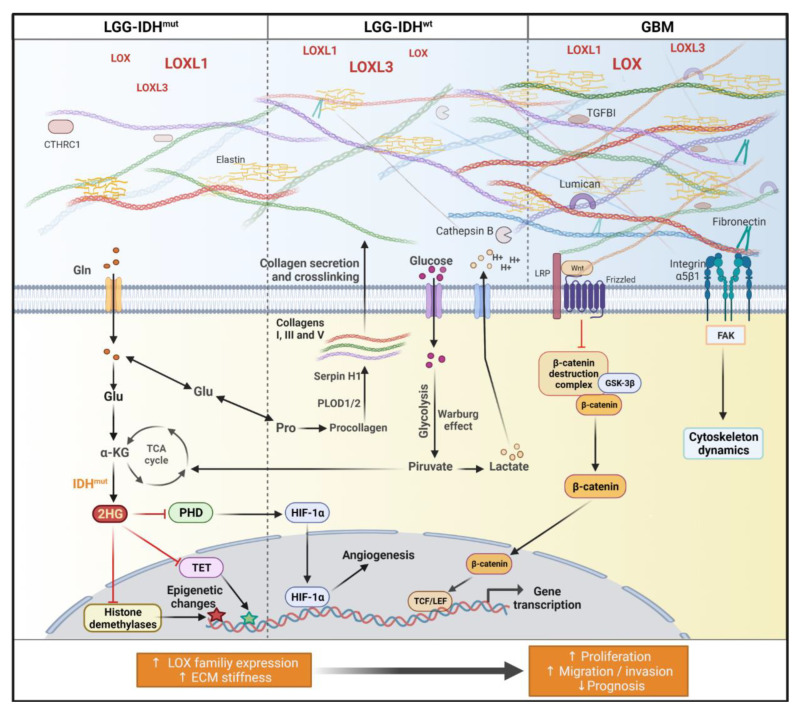
Schematic diagram representing LOX family influence in extracellular matrix composition of astrocytomas. Malignancy and aggressiveness increase from LGG-IDH^mut^, LGG-IDH^wt^, and through to GBM, and have been attributed to the IDH mutation status and ECM stiffness. ECM composition in turn was directly related to differential expression levels of *LOX*, *LOXL1*, and *LOXL3*, and their correlations with matrisome-associated genes, especially fibrillar collagens type I, III, and V. *LOXL1* expression level correlated significantly with expression of *CTHRC1* in LGG-IDH^mut^. In LGG-IDH^wt^, strong correlations were observed between *LOXL3* and cathepsin B expression levels. *LOX* expression correlated significantly with *PLOD1* and *PLOD2* (related to collagen biosynthesis) and with serpin H1 (involved in collagen folding). In IDH^wt^, Glu is accumulated and converted to Pro favoring collagen biosynthesis. The Warburg effect, with increased glucose uptake and glycolysis, leads to lactate production with lowering of extracellular pH, which promotes cathepsin B secretion and contributes to ECM degradation and remodeling. In GBM, *LOX* expression correlated with genes coding for lumican, TGFI, and fibronectin expression levels. ECM stiffness triggers mechanotransduction through α5β1 integrins and Wnt/β-catenin signaling with translocation of β-catenin to nucleus and transcription genes related to tumor malignant progression. Abbreviations: 2HG, 2-hydroxyglutarate; α-KG, α-ketoglutarate; CTHRC1, collagen triple helix repeat containing-1; ECM, extracellular matrix; FAK, focal adhesion kinase; GBM, glioblastoma; Gln, glycine; Glu, glutamine; GSK-3β, glycogen synthase kinase 3 beta; HIF-1α, hypoxia-inducible factor 1 alpha; IDH^mut^, isocitrate dehydrogenase gene with mutation; IDH^wt^, isocitrate dehydrogenase gene wild type; LEF, lymphoid enhancer factor; LGG, lower-grade astrocytic gliomas; PHD, prolyl hydroxylase; Pro, proline; TCA, tricarboxylic acid; TCF, T cell factor; TET, ten eleven translocation; Wnt, wingless and Int-1. Created with BioRender.com.

**Table 1 ijms-23-09507-t001:** Oligonucleotides for RT-qPCR reactions.

Gene	PCR Product (bp)	Orientation	Primer Sequences (5′-3′)
*LOX*	117	Sense Antisense	CCTACTACATCCAGGCGTCCA CATAATCTCTGACATCTGCCCTGT
*LOXL1*	162	Sense Antisense	GCTATGACACCTACAATGCGGA GACCTGTGTAGTGAATGTTGCATCT
*LOXL2*	112	Sense Antisense	ACCCACCCACTATGACCTGCT CTCGTAATTCTTCTGGATGTCTCCT
*LOXL3*	115	Sense Antisense	CTGGAACAGGCCGCATCT CCCCAGCATCCTCATCGT
*LOXL4*	115	Sense Antisense	GGCAGAGTCAGATTTCTCCAACA GAGTTCTGCATTGGCTGGGTAT
*GUSB*	101	Sense Antisense	GAAAATACGTGGTTGGAGAGCTCATT CCGAGTGAAGATCCCCTTTTTA
*HPRT*	118	Sense Antisense	TGAGGATTTGGAAAGGGTGT GAGCACACAGAGGGCTACAA
*TBP*	98	Sense Antisene	AGGATAAGAGAGCCACGAACCA CTTGCTGCCAGTCTGGACTG

**Table 2 ijms-23-09507-t002:** Primary antibodies used for immunohistochemistry.

Antibodies	Specificity	Company	Positive Control	Dilution
LOX	rabbit polyclonal	Abcam	Placenta	1:400
LOXL1	rabbit polyclonal	Sigma-Aldrich	Esophagus	1:100
LOXL3	rabbit polyclonal	Aviva	Placenta	1:50

## Data Availability

Not applicable.

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
