# Peer review of "Correlation of Matrisome-Associatted Gene Expressions with LOX Family Members in Astrocytomas Stratified by IDH Mutation Status"

_ijms, 2022, doi:10.3390/ijms23179507_

Round 1

Author Response

Dear reviewer,

We appreciated the comments and suggestion to improve the manuscript. We took into consideration all comments and we made the modification as suggested. We marked up all the modification in the manuscript using Track Changes.

The lysyl oxidases (LOXs) are a family of secreted copper-dependent enzymes are play role in facilitation extracellular matrix (ECM). The abnormal expression of these enzymes has various disease implications. Authors trying to dissect the LOX family enzyme expression in

astrocytomas especially in LGG- IDHmut, LGG-IDHwt and GBM groups.

Introduction is sufficient provides necessary background information.

Materials and methods explained well.

The results were explained to the point.

Except couple of typos “GBM patients (p = 0.032) (Figure 3B). 127 should be 3D”

In figure 4: N= 70 genes in last box showed N = 46?

Line 193 talks about fig 5C but there is no figure 5 C

The explanation for Figure 6B is very poor and very hard to understand from figure or through explanation.

Response: All points were changed according to suggestions. The explanation of Figure 6B was also improved.

Discussion is very hard fallow; it looks like the authors are trying to compensate for short

introduction. The discussion sounding like incoherent review for Matrisome analysis or trying to connect top candidates form the analysis to in very complex way to LOX.

Response: The discussion was considerably changed and focused on possible signaling pathways related to IDH mutation status and correlated with the expression levels of LOX family members and genes coding for matrisome components with impact on astrocytoma progression.

The figure 7 explains summary very well from results. I would request the authors reconstruct the discussion from Figure 7 more simpler way. That should improve the quality of discussion as well as publication quality.

Response: The legend of Figure 7 was restructured.

There are many casual English sentences in the paper and needs significant improvement in the paper

Response: Manuscript was submitted for second round of English review as suggested.

Reviewer 2 Report

The presented paper entitled "Impact of extracellular matrix stiffness by LOX family in astrocytoma progression" is out of interest.

The study is not related to matrix stiffness investigations. The authors only found the associations between mRNAsexpressions and proteins in tissues. These results are obvious. The other content of the paper is only speculating things. 

I recommend rejecting this paper.

Author Response

Dear reviewer,

We appreciated the comments about the manuscript.

Reviewer 3 Report

The manuscript “Impact of extracellular matrix stiffness by LOX family in astrocytoma progression” highlight the expression of LOX family genes in lower-grade astrocytoma and glioblastoma samples from a local cohort and further strengthened the findings by an extensive analysis from RNA-seq databases regarding LOX family expression in astrocytoma samples isocitrate dehydrogenase gene mutant and wild-type (IDH-mut and IDH-wt, respectively).

I’d like to congratulate the authors for the high-quality work.

While reading the manuscript, there were some questions and suggestions that I believe might be of interest to consider:

·         Figure 1: it might be interesting to add at the figure the number of samples in each group in figure A. It is well described in “material and methods” section, but I believe the interpretation of the data would be improved by making this information available in the figure.

·         Figure 2: Was the immunohistochemistry performed in the entire casuistic samples (130)? If not, please inform in the text the number of samples evaluated.

·         Figure 6: the correlation between CTNNB1 and LOXL1 might be related to LOXL1 promoter region having a cluster of five TCF/LEF elements in its transcription start site (PMID: 33981809) and it might be interesting to consider this information for this data interpretation. It might be possitble to speculate about a possible positive-feedback between CTNNB1-LOXL1.

  • Also, it would strengthen the data to evaluate the correlation between LOX family genes expression and overall survival in exclusively in patients submitted to radiotherapy (RT) in order to evaluate it LOX family genes are could be a predictor to RT responsiveness.

Author Response

Dear reviewer,

We appreciated the comments and suggestion to improve the manuscript. We took into consideration all comments and we made the modification as suggested. We marked up all the modification in the manuscript using Track Changes.

While reading the manuscript, there were some questions and suggestions that I believe might be of interest to consider:

Figure 1: it might be interesting to add at the figure the number of samples in each group in figure A. It is well described in “material and methods” section, but I believe the interpretation of the data would be improved by making this information available in the figure.

Response: The number of samples were added according to the suggestion.

Figure 2: Was the immunohistochemistry performed in the entire casuistic samples (130)? If not, please inform in the text the number of samples evaluated.

Response: The immunohistochemistry was performed in 6 cases of each AG2, AG3, and GBM. This information was added in Material and Methods.

Figure 6: the correlation between CTNNB1 and LOXL1 might be related to LOXL1 promoter region having a cluster of five TCF/LEF elements in its transcription start site (PMID: 33981809) and it might be interesting to consider this information for this data interpretation. It might be possible to speculate about a possible positive-feedback between CTNNB1-LOXL1.

Also, it would strengthen the data to evaluate the correlation between LOX family genes expression and overall survival in exclusively in patients submitted to radiotherapy (RT) in order to evaluate it LOX family genes are could be a predictor to RT responsiveness. OK

Response: The possible relationship of correlation between LOXL1 and CTNNB1 because of the presence of LOXL1 promoter region having a cluster of five TCF/LEF elements in its transcription start site was added in our discussion.

The analysis of overall survival exclusively in patients submitted to radiotherapy was performed according to the expression levels of the LOX family genes, and only the LOXL1 expression level impacted in the OS.  This result was added in this manuscript.

Round 2

Reviewer 1 Report

The authors responded well to review comments. I don't have any comments or suggestions!

Congratulations to the authors.